

# 1 Reflection tomography of time-lapse GPR data for studying
# 2 dynamic unsaturated flow phenomena

Adam R. Mangel[1,2], Stephen M.J. Moysey[2], John Bradford[1]
[1] Department of Geophysics, Colorado School of Mines, Golden, Colorado, 80401, USA
[2] Department of Environmental Engineering and Earth Science, Clemson University, Clemson, South Carolina, 29634,
USA
*Corresponding to*: Adam R. Mangel (amangel@mines.edu)
**Abstract**
Ground-penetrating radar (GPR) reflection tomography algorithms allow non-invasive monitoring of water content
changes resulting from flow in the vadose zone. The approach requires multi-offset GPR data that is traditionally
slow to collect. We automate GPR data collection to reduce the survey time significantly, thereby making this
approach to hydrologic monitoring feasible. The method was evaluated using numerical simulations and laboratory
experiments that suggest reflection tomography can provide water content estimates to within 5-10% vol./vol. for the
synthetic studies, whereas the empirical estimates were typically within 5-15% of measurements from in-situ probes.
Both studies show larger observed errors in water content near the periphery of the wetting front, beyond which
additional reflectors were not present to provide data coverage. Overall, coupling automated GPR data collection with
reflection tomography provides a new method for informing models of subsurface hydrologic processes and a new
method for determining transient 2D soil moisture distributions.

## 20 1. Introduction

21        Preferential flow is ubiquitous in the vadose zone, occurring under a wide variety of conditions and over a

broad range of scales (Nimmo, 2012). Reviews such as those by Hendrickx and Flury (2001) and Jarvis (2007)
illustrate that a basic mechanistic understanding of preferential flow exists. Jarvis et al. (2016) point out, however,
that we still lack models capable of reproducing empirical observations in the field and highlight the importance of
non-invasive imaging techniques for improving this understanding. We suggest that ground-penetrating radar (GPR)
reflection tomography could fill this need by quantitatively mapping changes in water content through space and time
at the sub-meter scale.

Reflection GPR is commonly used to image subsurface structures, but is also well suited to understanding

hydrologic variability due to the strong dependence of EM wave velocities on soil volumetric water content (Topp et
al., 1980). As a result, GPR has been adapted to monitor variability in hydrologic processes at multiple scales through
time and space in a variety of contexts (Buchner et al., 2011; Busch et al., 2013; Guo et al., 2014; Haarder et al., 2011;
Lunt et al., 2005; Mangel et al., 2012, 2015b, 2017; Moysey, 2010; Saintenoy et al., 2007; Steelman and Endres, 2010;
Vellidis et al., 1990)

While these studies have illustrated a variety of techniques for monitoring changes in water content within

the subsurface, they have generally required assumptions to constrain the interpretation, such as the use of *a priori*
information regarding subsurface structure (e.g., Lunt et al., 2005) or the GPR wave velocity (Haarder et al., 2011).



These limitations arise from the fact that GPR data are recorded as energy arriving at the receiver antenna as a function
of time. Inherent assumptions therefore exist in analyzing traveltime data collected with antennas separated by a fixed
offset because both the distance travelled by the GPR wave to a reflector and the velocity of the GPR wave are
unknown. It has been demonstrated that GPR data collected via a multi-offset survey can constrain both the depth to
a moving wetting front and the water content behind the front over the course of an infiltration event (Gerhards et al.,
2008; Mangel et al., 2012). The limitation of these studies, however, was that the authors assumed a 1D flow system
and that GPR data lacked information regarding lateral variability in soil moisture.
Extending multi-offset techniques to image flow in the vadose zone requires an increase in the speed at which
these data can be collected and advanced processing methods that can combine thousands of measurements into
spatially and temporally variable water content estimates. We have recently overcome the data collection problem by
automating GPR data collection using a computer controlled gantry, thereby reducing the data collection time for large
multi-offset surveys from hours to minutes (Mangel et al., 2015a). Reflection tomography algorithms have been
available in the seismic industry for decades (Stork, 1992) and were first adapted to imaging GPR velocity variations
by Bradford (2006). Subsequent studies have demonstrated the use of GPR reflection tomography for imaging static
distributions of subsurface water content with great detail (Bradford, 2008; Bradford et al., 2009; Brosten et al., 2009).
The combination of automated GPR data collection and reflection tomography makes time-lapse imaging of water
content during infiltration a feasible means to study flow in the vadose zone.
The objective of this study is to evaluate reflection tomography of high-resolution GPR data as a tool for
observing and characterizing unsaturated flow patterns during infiltration into a homogeneous soil. To evaluate the
efficacy of the algorithm for resolving dynamic soil water content in 2D, we first test the algorithm using numerical
simulations and compare the results to true water content distributions. We then apply the algorithm to time-lapse
GPR data collected during an infiltration and recovery event in a homogeneous soil and compare results to
measurements from in-situ soil moisture probes.

## 2.  Methods

### 2.1.  The Reflection Tomography Algorithm

The goal of reflection tomography is to determine a velocity model that best aligns migrated reflection
arrivals for a common reflection point across a set of source-receiver offsets. For brevity, we will limit our discussion
here to the key ideas and methods of the tomography algorithm; we refer the reader to Stork (1992) for the original
tomography algorithm and to Bradford (2006) for the application to GPR data.
The data required for this algorithm are an ensemble of common-midpoint (CMP) gathers collected along a
path. Given that GPR data is a time-series record of electromagnetic energy arriving at a point in space, we must
know the proper velocity structure to migrate the data and produce a depth registered image of the GPR energy.
Migration attempts to remove the hyperbolic trend of reflections with respect to antenna offset (Figure 1a) by using
the wave velocity to reposition reflections to the proper depth at which they occur. If CMP data are migrated with the
correct velocity, reflections from layers in the subsurface are flattened as a function of offset (Fig. 1c). If the velocity
estimate is incorrect, e.g. 10% too slow (Fig. 1b) or 10% too fast (Fig. 1d), the arrival is not flat and exhibits residual



moveout (RMO). To solve for the velocity structure and properly migrate the data, the reflection tomography
algorithm proceeds as follows [*Bradford* 2006; *Stork* 1992]:

1. Generate a starting depth vs. velocity model.
2. Migrate the data with the starting velocity model and stack the data.
3. Pick horizons on the stacked image.
4. Perform ray-tracing to the picked horizons with the velocity model.
5. Evaluate horizons for residual moveout.
6. Adjust velocity model using reflection tomography.
7. Apply revised velocity model using migration and quality check RMO.
8. Iterate at step three if necessary.

For this work, starting velocity models for the tomography algorithm are determined by smoothing results from 1D velocity analysis of individual CMPs (Neidell and Taner, 1971). The reflection tomography algorithm then adjusts the velocity distribution until reflections in the depth corrected (i.e., migrated) data line up to produce a reflection at a consistent depth across all traces in a CMP. Through sequential iterations of the tomographic inversion, the RMO metric is reduced on a global scale. For this work, the reflection tomography was performed using the SeisWorks software suite and Kirchhoff pre-stack depth migration (Yilmaz and Doherty, 2001).

**2.2. Experimental Setup and Procedure**

We used a 4 m x 4 m x 2 m tank for the controlled study of unsaturated flow phenomenon with GPR (Fig. 1e, f). We filled the tank with a medium-grained sand to a depth of 0.60 m. Below the sand was a 0.30 m layer of gravel that acts as backfill for 16 individual drain cells that are pitched slightly toward central drains that route water to outlets on the outside of the tank. We constructed an automated data collection system to allow for the high-speed high-resolution collection of GPR data (Mangel et al., 2015a); the GPR gantry fits inside of the tank so the antennas are in contact with the sand surface. All GPR data described here were collected along the y-axis of the tank at a fixed position of x = 2.0 m, where the bottom of the tank is flat (Fig. 1e, f).

The automated system, which utilizes a 1000 MHz Sensors and Software bistatic radar (Sensors and Software, Inc.), was operated to obtain 101 CMPs spaced at 0.02 m intervals between y = 1.0 - 3.0 m. Each CMP consisted of 84 traces with offsets between 0.16-1.0 m at 0.01 m step size. Thus, a complete CMP data set for one observation time consists of almost 8,500 individual GPR traces. With this configuration using the automated system, a CMP at a single location could be collected in 1.8 seconds with a total cycle of CMP data locations collected every 3.9 minutes.

GPR data collection occurred prior to irrigation to evaluate background conditions. Data collection continued during irrigation, which was applied at a flux of 0.125 cm/min for a duration of 2.13 hrs. Spatial heterogeneity in the applied flux has been observed in laboratory testing of the irrigation equipment. Fifteen EC-5 soil moisture probes (METER, Inc.) logged volumetric water content at 10 second intervals during the experiment (Fig. 1e, f). Note that the soil moisture probes are located out of the plane of the GPR line by 0.5 m (Figure 1f). GPR data collection continued for 40 min. after the irrigation was terminated. In total, 45 complete sets of data were collected over the course of the 3-hour experiment, yielding more than 500,000 GPR traces in the experimental data set.





### 2.3. Execution of the Numerical Simulations


We employed HYDRUS-2D (Simunek and van Genuchten, 2005) to simulate a theoretical and realistic
hydrologic response to an infiltration event using two different initial conditions: i) hydrostatic equilibrium leading to
a water content distribution controlled by the soil water retention curve, and ii) a uniform soil with a water content of
0.07. We selected the Mualem-van Genuchten soil model (Mualem, 1976) and parameterized the model as follows
based on hydraulic testing of the sand: residual water content ($\theta_r$) = 0.06, saturated water content ($\theta_s$) = 0.38, air-entry
pressure ($\alpha$) = 0.058 cm$^{-1}$, shape parameter ($n$) = 4.09, and saturated hydraulic conductivity ($K_s$) = 4.6 cm min$^{-1}$. The
hydraulic conductivity for the homogeneous model was reduced to 1 cm min$^{-1}$ to build a larger contrast of water
content across the wetting front. For all HYDRUS simulations, we used a constant flux boundary condition of 0.125
cm/min from y = 1.6 - 2.4 m along the ground surface, set the model domain depth to 0.6 m, length to 4.0 m, and
nominal cell size to 0.04 m. Remaining nodes at the surface were set to no flow boundaries and lower boundary nodes
were set to a seepage face with the pressure head equal to zero.
We calculated relative dielectric permittivity values for the GPR simulations by transforming water content values
from HYDRUS-2D using the Topp equation (Topp et al., 1980). We used the magnetic permeability of free space for
the entire model domain and set electrical conductivity of the soil to 1 mS/m. Although electrical conductivity changes
as a function of the water content, these changes primarily influence wave attenuation, which is not significant or
accounted for in the processing performed with the SeisWorks software.
We performed GPR simulations in MATLAB using a 2D finite-difference time-domain code (Irving and
Knight, 2006). The GPR model domain was set to 4.0 m long and 1.1 m high with a cell size of 0.002 m. The lower
0.3 m of the domain was set to a relative dielectric permittivity of 2.25 to represent the lower gravel layer and the
upper 0.2 m was modeled as air to simulate the air-soil interface. Simulated data were collected as described in the
section detailing the tank experiment. For quick computation, simulations were deployed on the Palmetto
supercomputer cluster at Clemson University, where single source simulations ran in 20 minutes using nodes with 8
CPUs and 32 GB of RAM.

### 3. Reflection TOmography Of simulations


The HYDRUS-2D output shows the development of an infiltrating wetting front for the two scenarios with
differing initial conditions (Figs. 2a, f, k). For conditions prior to irrigation, the bottom of sand reflection (B) is
horizontal on the common-offset profile (COP) data indicating a constant velocity across the model domain (Fig. 2b).
Additionally, the CMPs show identical hyperbolic moveout, i.e., the offset vs. traveltime relationship, indicating a
homogeneous velocity across the model domain (Fig. 2c-e). The airwave and groundwave are also visible in the data,
but are not analyzed, or further discussed.
During infiltration, (B) is distorted at the center of the COP due to the decreased velocity caused by the
infiltrating water (Figs. 2g, l). A reflection from the infiltrating wetting front (W) is faintly visible for the model with
variable initial water contents (Fig. 2g) and comparatively strong for simulations with a dry background (Fig. 2l) due
to different levels of dielectric contrast across the wetting front in each case. CMPs also indicate perturbations in the
velocity field as the moveout changes dramatically when the wetted zone is encountered (Figs. 2h-j, m-o). A refraction



is also observed on the CMPs, which is a rare occurrence considering that GPR wave velocity typically decreases with
depth.

Prior to the onset of flow, the reflection tomography algorithm produces a uniform water content distribution

that agrees with the arithmetic average of the true water content but does not capture the vertical gradation observed
in Figure 3a. This is because information regarding vertical velocity variations is absent, i.e., more reflectors at
different depths are required to capture this variability. As a result, errors in the water content estimation exceed 10%
vol./vol (Fig. 3c).

During infiltration the wetting front is imaged relatively well for the case where the soil was initially dry

(Figs. 3g-i), particularly as the plume advances deeper into the subsurface (Figs. 3j-l) where there is improved data
coverage. Considerable errors in the tomography results persist, however, with the results degrading further for the
scenario with variable initial water content (Figs. 3d-f) given that reflection contrasts with the wetting front are weaker.
The presence of an additional reflector, however, increases the ability of the tomography to resolve vertical variability,
e.g. Figure 2e vs. Figure 2b. Overall, errors are reduced near reflectors to about 5% vol./vol. These results suggest
that water content changes resulting from unsaturated flow can be imaged and that as more information becomes
available in the form of additional reflections, the tomography results improve.

## 4.    Reflection tomography of Experimental data

At initial conditions, the sand layer reflection (B) is visible at 10 ns traveltime in the COP collected over the

imaging area (Fig. 4a). Normal hyperbolic moveout of (B) is observed on the CMPs (Fig.4b, c, d). These results are
qualitatively identical to observations from numerical simulations (Figs. 2b-e).

During infiltration, the water content of the sand layer increases substantially (Fig. 5) and longer traveltimes

of the arrivals on the COP data are observed near the center of the tank (Figs. 4f, i). Rather than a coherent reflection
for the wetting front (W) (Fig. 2l), multiple discrete reflections are present in the COP data (Fig. 4e, i, m) indicating
a heterogeneous wetting of the soil. These reflections are difficult to identify on the CMPs given the complex moveout
pattern (Fig. 4i) but are more easily identified in animations of COP projections of the data (included as a
supplementary file). Analysis of the data was greatly aided by the animation of the data and the pre-stack migration
algorithm, which stacks the data over all offsets to produce a coherent image of reflectors with an increased signal to
noise ratio. Heterogeneous wetting of the soil is also very pronounced in the soil-moisture probe data with many of
the probes responding out of sequence with depth (Fig 5). After irrigation, the soil moisture probes show a decrease
in the soil water content (Fig. 5) apart from one probe (Fig. 5c) and the GPR data show a slight decrease in the
traveltime of the bottom of sand reflection (Figs. 4k-n).

The tomographic imaging results from the initial GPR data set collected prior to irrigation agree with data

from soil moisture probes which indicates an average soil moisture of roughly 5% during this time (Figs. 4e, 5).
During infiltration and recovery, tomographic images of the tank show a wet zone at the center and relatively dry
edges outside the irrigated area (Figs. 4j, o). Overall, the tomography results near the center of the tank are within
10% vol./vol. of the soil moisture data and show a non-uniform wetting of the soil during infiltration that was not
observed in the numerical study, suggesting the occurrence of preferential flow. Errors in the estimates of water
content near the edges of the advancing plume exceed 15% vol./vol. (Fig. 4b, c), though the general patterns in wetting



are consistent. After irrigation, the tomography results on the edges of the wetted zone are in better agreement with
the soil moisture probe data, but less spatial information is available given the lack of a wetting front reflection (Fig.
4o).
**5. Conclusions**

Reflection tomography in the post-migrated domain is a viable method for resolving transient soil moisture

content in 2D associated with an infiltration and recovery event in a homogeneous soil. Reflection tomography of
numerical data produced water content distributions that were in good agreement with true water content values from
the simulations. The tomography was generally able to match the true water content values to within 5-10% vol. /vol.
However, distinct migration artifacts were produced around the edges of the wetting front, especially for cases where
the initial water content was non-uniform, obscuring details about the shape of the wetted region. Analysis of data
collected in a sand tank proved to be more difficult, however, the tomography was able to produce hydrologically
realistic distributions of water content in space and time that were generally within 5-15% vol./vol. of measurements
from in-situ soil moisture probes. This may have to do with the complex distribution of the wetted soil as a result of
heterogenous distribution of water at the surface, texture variability in the soil, or other preferential flow mechanisms
(Jarvis et al., 2016). Regardless, the fact that the GPR data were able to capture this heterogeneity is an impressive
feat given that tomographic imaging operated independently of any hydrologic information and provided evidence
that our conceptual model was not representative of the physical system.

Regardless of discrepancies observed between the GPR and probe water content values, it is evident that

automated high-speed GPR data acquisition coupled with reflection tomography algorithms can provide a new
approach to hydrologic monitoring. Testing and revision of conceptual hydrologic models regarding non-uniform
flow in the vadose zone demands such non-invasive time-lapse imaging data. Artifacts observed in the numerical
simulation results, however, suggest that improvements in this methodology could be achieved. While there are likely
fundamental limitations to the information content of the data, the Kirchhoff pre-stack depth migration algorithm used
in this study could be replaced by more sophisticated algorithms like reverse-time migration (Baysal et al., 1983)
which may reduce the observed imaging artifacts. Additionally, results from the tomography algorithm may prove to
be beneficial as a precursor to higher-order inversion techniques, like full-waveform inversion, which requires detailed
starting models of velocity for convergence. Overall, coupling automated GPR data collection with reflection
tomography provides a new method for informing models of subsurface hydrologic processes and a new method for
determining transient 2D soil moisture distributions.
**6. Acknowledgements**

This material is based upon work supported by, or in part by, the National Science Foundation under grant

number EAR-1151294. We also acknowledge Clemson University for generous allotment of compute time on
Palmetto cluster. Data used in this publication and a supplementary movie of the data are available through the
Colorado School of Mines at the following URL: https://hdl.handle.net/11124/172053.





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

**Figure 1**

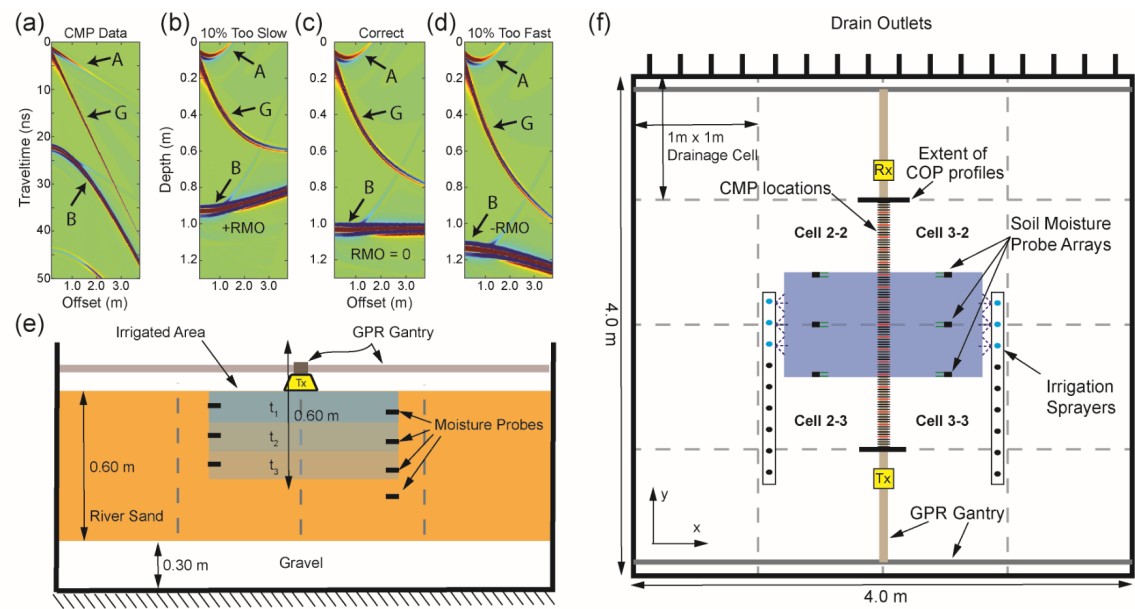


Figure 1: a) Example CMP data showing the airwave (A), groundwave (G) and reflection from a layer (B). Data in (a) is migrated to form (b) a migrated gather
with velocity 10% slow; c) a migrated gather with correct velocity; and d) a migrated gather with velocity 10% fast. Panel (e) shows a cross-section of the experiment
at y = 2.0 where $t_1$, $t_2$, and $t_3$ are arbitrary times during the infiltration. Panel (f) shows the plan-view of the experiment. Note that the bottom of the sand layer is
flat where GPR data collection occurs, i.e. on a boundary between drain cells, and pitched elsewhere toward cell drains.




**Figure 2**

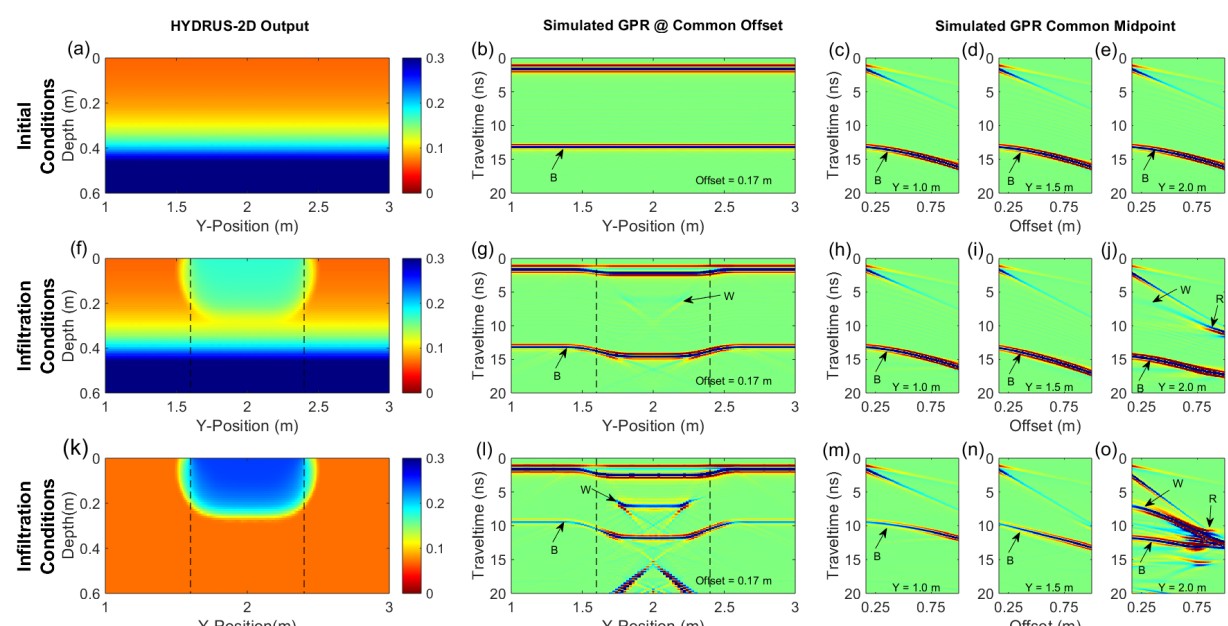


Figure 2: Panels (a), (f), and (k) show volumetric moisture distribution from HYDRUS-2D simulations used to generate simulated common-offset GPR data (b, g,
l) and multi-offset GPR data (c-e, h-j, and m-o). Vertical dashed lines indicate the extent of the wetted surface. Annotated arrivals are the bottom of sand layer
reflection (B), wetting front reflection (W), and refraction (R). Note that the base of sand reflection (B) is caused by the boundary at 0.60 m depth between the sand
and gravel, not the capillary rise shown in panels (a) and (f).

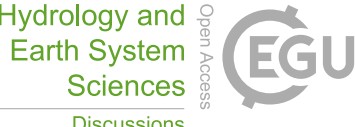



**Figure 3**

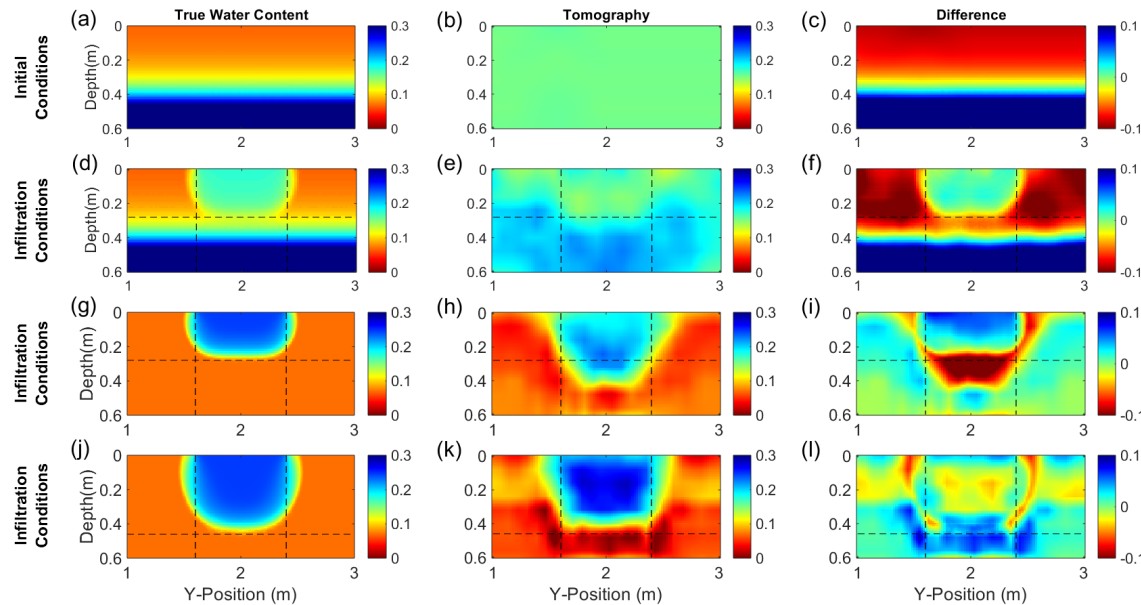

Figure 3: Panels (a), (d), (g), and (j) show true volumetric water content distributions from HYDRUS-2D. Panels (b), (e), (h), and (k) show results of tomography of the simulated GPR data as volumetric water content. Difference plots (c), (f), (i), and (l) were calculated by subtracting the tomography results from the true water content distributions; red areas indicate volumetric moisture underestimation while blue areas indicate volumetric moisture overestimation.





**Figure 4**

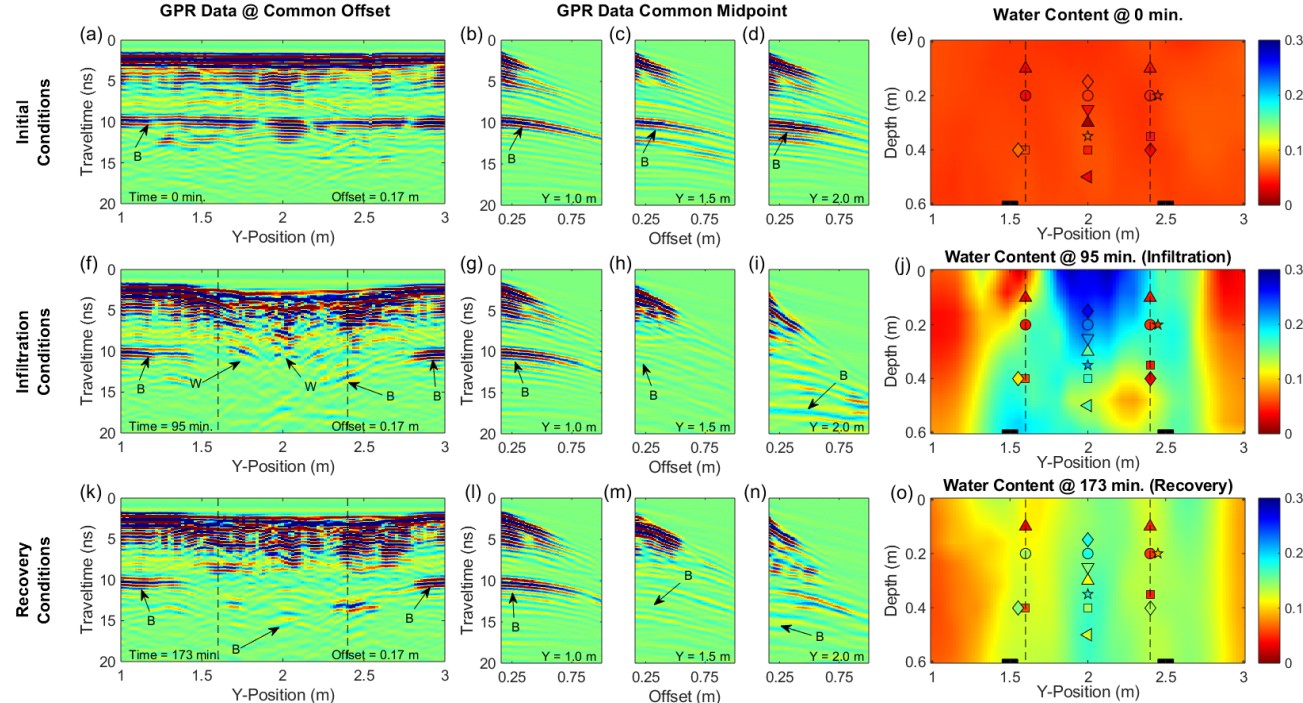


Figure 4: Panels (a, f, and k) are common-offset GPR data collected during the experiment. Panels (b-d, g-i, and l-n) are CMP data collected during the experiment.
Arrivals annotated are the sand layer reflection (B) and wetting front reflection (W). Panels (e, j, and o) show tomography results for the corresponding GPR data.
Vertical lines indicate the lateral extent of the wetted surface. Shapes correspond to the soil moisture data for the given y-location in Figure 5, colors correspond to
the measured soil moisture. Adjacent symbols are from probes that are located at different x-locations, but identical depths.





**Figure 5**

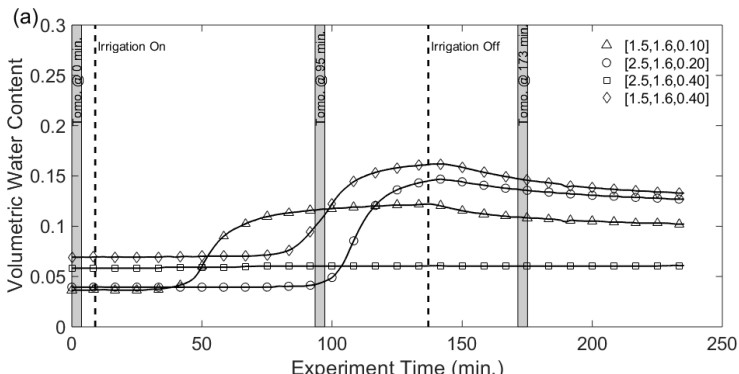

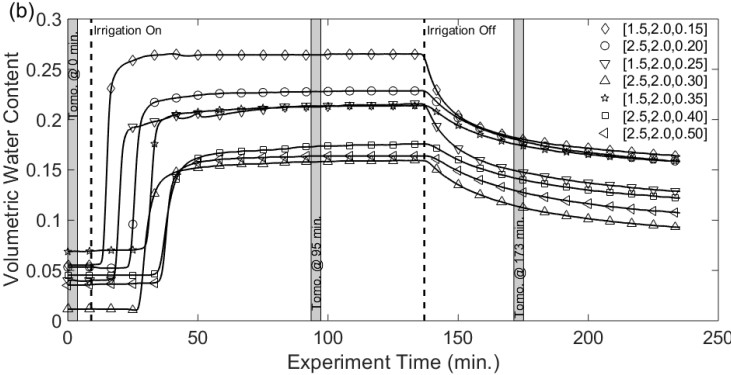

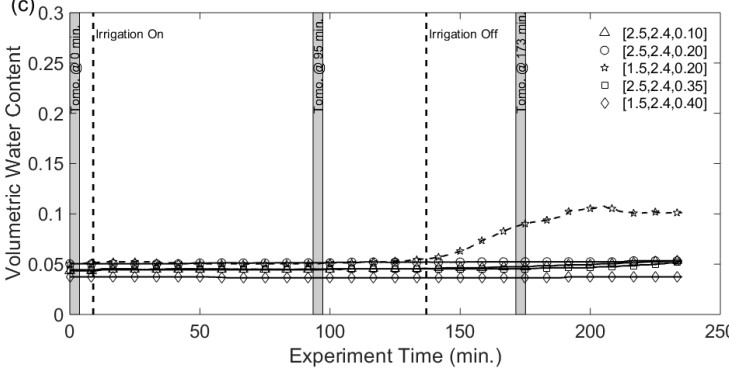


Figure 5: Soil moisture probe data from the in-situ moisture probes along the GPR line at a) y = 1.6 m; b) y = 2.0 m;
and c) y = 2.4 m.  Vertical dashed lines indicate the start and stop of irrigation. Gray bars indicate the times when data
in Figure 4 were collected.  Symbols for a given data set match those on Figures 4e, j, and o. Soil moisture data were
collected 60 minutes beyond the end of GPR data collection.