# Peer review of "Reflection tomography of time-lapse GPR data for studying"

_Hydrology and Earth System Sciences, 2018_

## Referee Comment (RC1) · Anonymous Referee #1 · 5 Aug 2018

I have found the paper interesting. And as such, with an application to the attempt of estimating moisture content from multi-offset GPR data, is largely novel. Yet, I have found it to a large extent disappointing that the authors seem to give too much credit to their own past work, and neglect a large body of literature considering multi-offset GPR processing, that dates back at least a couple of decases. Even more serious, is the lack of proper reference to wave migration methods that are state of the art in industrial seismic processing, and are yet presented herewith as if they are novel, or at least rediscovered by the authors. The reference list is poor in both respects, but particularly with concern to migration algorithms (only Stork 1992 is mentioned, that dates back some 25 years, and later Yilmaz and Doherty, 2001). I encourage the authors

to widen the literature review and put their work, not without merits, in the correct perspective. See below some suggestions for references to be put in the correct context, particulary in terms of GPR applications (but not only). From a technical viewpoint, I am a bit puzzled by the error estimates for water content estimates that is 5-10% in vol/vol (is it saturation or moisture content?) – as compared to 5-15% from soil probes (again, same question). I feel this error is far too high to make the estimates useful (if it is moisture content as I read it!). Note that in cross-hole GPR usually a 2-3% error in volumetric moisture content is generally accepted as realistic. Finally, as much as I like GPR, it should be clearly stated in the introduction that GPR can only be used in relatively resistive soil conditions. This is generally omitted when presenting GPR applications, yet in many practical situations the soil conductivity is high enough to force us to shift to ERT or EMI for soil moisture content estimates. References Forte E. and M. Pipan, 2016, Review of multi-offset GPR applications: Data acquisition, processing and analysis, Signal Processing, doi: 10.1016/j.sigpro.2016.04.011 Jaumann S. and K. Roth, 2018, Soil hydraulic material properties and layered architecture from time-lapse GPR, Hydrol. Earth Syst. Sci., 22, 2551–2573, doi: 10.5194/hess-22-2551-2018 Lambot, S., Antoine, M., Van den Bosch, I., Slob, E., and Vanclooster, M.: Electromagnetic inversion of GPR signals and subsequent hydrodynamic inversion to estimate effective vadose zone hydraulic properties, Vadose Zone J., 3, 1072–1081, https://doi.org/10.2113/3.4.1072, 2004. Lambot, S., Slob, E., Rhebergen, J., Lopera, O., Jadoon, K. Z., and Vereecken, H.: Remote estimation of the hydraulic properties of a sand using full-waveform integrated hydrogeophysical inversion of time-lapse, offground GPR data, Vadose Zone J., 8, 743–754, https://doi.org/10.2136/vzj2008.0058, 2009. Leparoux D., D. Gibert, P. Côte, 2001, Adaptation of prestack migration to multi‐offset ground‐penetrating radar (GPR) data, Geophysical prospecting, 49(3), 374-386, doi: 10.1046/j.1365-2478.2001.00258.x Klenk, P., Jaumann, S., and Roth, K.: Quantitative high-resolution observations of soil water dynamics in a complicated architecture using time-lapse ground-penetrating radar, Hydrol. Earth Syst. Sci., 19, 1125–1139, doi: 10.5194/hess-19-1125-2015, 2015.

---

## Referee Comment (RC2) · Anonymous Referee #2 · 23 Oct 2018

In general, I found the paper interesting. It is well written, and you can follow the methodology of the investigators. The way the experiments were carried out, specially the automated data collection system is interesting. The numerical analysis adds value to the paper. However, there are some points that needs to be addressed and clarified. I would like the authors respond to the following questions and clearly explain their ideas and points of views:

1) In section 2.2, line 128, authors mention that 101 CMPs were collected (between y=1m and y=3m). I think there must be a typo here. It should "COPs." Otherwise, it does not make sense. If the transmitter and the receiver have moved 2cm each

time, you should have 51 profiles. If each one has moved 1cm at a time, then you will have 101 profiles. Please fix and/or clarify. 2) It is mentioned that the flux is 0.125 cm/min, but the authors did not explain how uniform the irrigation was (inside the irrigation area). 3) The moisture probes were situated 0.5 m away from the line of GPR scan. Unless, the irrigation was uniform, it does not make sense to compare the results of moisture content from the GPR scans to moisture content data from the probes. I guess, we assume the sand layer was homogeneous. 4) In line 183, it is mentioned that a refraction is also observed on the CMPs. Please discuss and explain why refraction happened in this case. 5) In section 4, line 225, the error was reported for water content near the edges of the advancing plume. Please explain how the errors were calculated considering the fact that the GPR scans were collected at fixed x=2.0 m and the probes are 0.5 m away from the line of scan. How did you calculate the water content error for the central area versus the edges of the plume. Please explain.

---

## Author Comment (AC1) · 31 Oct 2018

This response is directed toward the comments left by Anonymous Referee #1, posted to the Hydrology and Earth Systems Sciences (HESS) discussion board for manuscript **hess-2018-230** on August 5, 2018.

First and foremost, the authors would like to thank Anonymous Referee #1 (AR1) for devoting time to reviewing our manuscript and for providing a critical review of the content. Below, the authors have outlined responses to individual comments made by the reviewer.

*AR1: Yet, I have found it to a large extent disappointing that the authors seem to give too much credit to their own past work, and neglect a large body of literature considering multi-offset GPR processing, that dates back at least a couple of decades.*

**Author Response**: The authors have added additional references for processing of multi-offset GPR data as suggested by the reviewer.

*AR1: Even more serious, is the lack of proper reference to wave migration methods that are state of the art in industrial seismic processing, and are yet presented herewith as if they are novel, or at least rediscovered by the authors.*

**Author Response**: The authors have added additional references for wave migration methods as suggested by the reviewer. The authors do not claim to have 'rediscovered' the methods used but are simply the first to apply them to time-lapse GPR data for imaging dynamic hydrologic conditions, which is novel.

*AR1: From a technical viewpoint, I am a bit puzzled by the error estimates for water content estimates that is 5-10% in vol/vol (is it saturation or moisture content?) – as compared to 5-15% from soil moisture probes (again, same question). I feel this error is too high to make the estimates useful (if it is moisture content as I read it!). Note that in cross-hole GPR usually 2-3% error in volumetric moisture content is generally accepted as realistic.*

**Author Response**: All soil moisture data is reported as volumetric water content. Errors calculated in the manuscript are absolute errors in volumetric water content between the soil moisture probe data and the values derived from analysis of the GPR data.

Errors in the estimation of volumetric water content from the simulated and measured GPR data vary over space and time. The authors offer explanations for these errors which will be the focus of later research, e.g. evidence of non-uniform wetting of the sand from soil moisture probe data and migration artifacts present in the Kirchoff migration. The authors do not recommend directly comparing results of this method to those of cross-hole GPR given increased data coverage using cross-hole GPR methods. The appeal of this method is partially because boreholes are not required to image the subsurface.

*AR1: Finally, as much as I like GPR, it should be clearly stated in the introduction that GPR can only be used in relatively resistive soil conditions. This is generally omitted when presenting GPR applications, yet in many practical situations the soil conductivity is high enough to force us to shift to ERT or EMI for soil moisture content estimates.*

**Author Response**: The authors have added a statement to the introduction that describes the limitations of GPR in conductive media.

---

## Author Comment (AC2) · 31 Oct 2018

This response is directed toward the comments left by Anonymous Referee #2, posted to the Hydrology and Earth Systems Sciences (HESS) discussion board for manuscript **hess-2018-230** on October 23, 2018.

First and foremost, the authors would like to thank Anonymous Referee #2 (AR2) for devoting time to reviewing our manuscript and for providing a critical review of the content. Below, the authors have outlined responses to individual comments made by the reviewer.

*AR2: In section 2.2, line 128, authors mention that 101 CMPs were collected (between y=1m and y=3m). I think there must be a typo here. It should "COPs." Otherwise, it does not make sense. If the transmitter and the receiver have moved 2cm each time, you should have 51 profiles. If each one has moved 1cm at a time, then you will have 101 profiles. Please fix and/or clarify.*

**Author Response**: The discussion of the data collection geometry in the manuscript has been reviewed by the corresponding author and is correct. A common-midpoint profile (CMP) was collected at 101 individual points between y = 1 m and y = 3 m (see Figure 1f in the manuscript). To collect a CMP, the transmitter and receiver are expanded or contracted about a central point. The central point locations are spaced at 2 cm intervals along the line at x = 2 m over the distance of y = 1 to 3 m. Therefore, the total transect distance of 2 meters (3 m – 1 m) is covered in 100 steps of 2 cm increments. Counting the starting position, this calculates to a total of 101 CMPs.

Of course, you could rearrange the data to give you multiple common offset profiles across the tank. The number of COPs in this case would depend on the number of offsets collected in each CMP, which for this experiment was 84. These COPs, however, would be collected with significant temporal disparity such that the traces at either end of the COP would be separated in time by 3.9 minutes. This would distort the reflectors in the COPs due to the dynamic nature of the infiltration process being monitored. The COPs are of little value to our work, however, as they do not contain the reflector moveout relationships that are used to estimate wave velocities. Thus, we only collected data in CMP configurations.

*AR2: It is mentioned that the flux is 0.125 cm/min, but the authors did not explain how uniform the irrigation was (inside the irrigation area).*

**Author Response**: Lines 104-105 of the manuscript mention heterogeneity in the applied flux. This may indeed be a factor in the heterogeneity of the wetting front, but we have gone through many iterations of the irrigation equipment to minimize the problem and have made progress in homogenizing the applied flux over a prescribed area. While we continue to work on this issue, it does not impact the validity of our results give that we do not fundamentally assume homogeneity of the infiltration flux across the tank for reflection tomography to be applicable. To the contrary, our GPR and moisture probe results are both consistent with some degree of variability occurring in the applied flux. For this work, we feel that homogeneous application of water at the surface is less important than the fact that we are able to discriminate a heterogeneous response, which is more representative of real systems and points to the capability of reflection tomography for measuring heterogeneous water distributions in the environment.

*AR2: The moisture probes were situated 0.5 m away from the line of GPR scan. Unless, the irrigation was uniform, it does not make sense to compare the results of moisture content from the GPR scans to moisture content data from the probes. I guess, we assume the sand layer was homogeneous.*

**Author Response**: The sand layer is homogeneous, meaning that it is all the same sand, from the same company, and was packed into the tank in a uniform manner. However, homogenous systems can exhibit heterogeneous flow responses due to small variabilities in initial and boundary conditions or very minor contrasts in sorting at the grain scale. The overall agreement between the patterns of water content change observed in the reflection tomography and probe responses suggest that the general comparison between the datasets we performed is valid, though we agree with the reviewer that over analysis of the two data sets is not warranted given this limiting factor as well as the fact that the measurements represent different scales of investigation.

The moisture probes were located off the GPR transect to avoid backscattering or the transmitted signal that would interfere with analysis of the GPR wave velocities. Though this is not an ideal setup for comparison of the moisture probe data to the GPR estimated of water content, it was necessary to ensure that high-quality reflection tomography data could be collected.

*AR2: In line 183, it is mentioned that a refraction is also observed on the CMPs. Please discuss and explain why refraction happened in this case.*

**Author Response**: Refraction occurs in this case because a wet low-velocity layer is present above a dry high-velocity layer. Overall, the refraction is irrelevant to this work, but the authors point it out as a rarity in the simulated GPR data. The refraction is not observed in the empirical GPR data and warrants no further discussion.

*AR2: In section 4, line 225, the error was reported for water content near the edges of the advancing plume. Please explain how the errors were calculated considering the fact that the GPR scans were collected at fixed x=2.0 m and the probes are 0.5 m away from the line of scan. How did you calculate the water content error for the central area versus the edges of the plume. Please explain.*

**Author Response**: Errors in the estimates of water content are calculated by comparing the difference between the soil moisture probe data and the values derived from analysis of the GPR data. We must assume homogeneity in the x-direction to directly compare these measurements which is why the errors are not discussed in greater detail in this work. Rather, the authors provide these numbers as a general metric regarding the performance of the GPR data analysis.

---

## Referee Report (RR1)

Review in support of the editor's decision for

**Mangel et al: Reflection tomography of time-lapse GPR data for studying dynamic unsaturated flow phenomena**

The paper under review for publication in HESS by Mangel et al aims at employing a reflection tomography based inversion algorithm, which is well-established for calculating subsurface velocity distributions from CMP GPR measurements in stationary conditions for deriving – by proxy – subsurface water content – distributions. In contrast to previous publications, here the focus is on dynamically changing conditions during infiltration experiments.

First of all, I would like to specifically laud the authors for their dedicated experimental approach and congratulate them for their laboratory setup and the undoubtedly involved data set which may yet hold the key to studying the infiltration experiments they monitored by GPR in so much detail.

However, the key question for whether the currently submitted work warrants a dedicated publication is whether the authors found a novel and robust way to extract meaningful and relevant information from this great dataset. Unfortunately, I am convinced that the inversion approach chosen for this publication in its current form falls short of achieving that aim (i.e., as the title states: Usage of this algorithm for "studying dynamic unsaturated flow phenomena") and does not give justice to the information potentially contained in their elaborated dataset.

The inversion algorithm's trouble is quite clearly shown already by the simulation based results the authors present in Figure 3: Here, the authors first calculate water content distributions from HYDRUS-2D (figure 3, left column), then derive GPR profiles from these distributions (examples shown in figure 2) and feed these into their tomography algorithm to retrieve the respective water content distributions (figure 3, center column). In the first case, as the authors admit themselves, their algorithm fails completely to capture the velocity profile, since there is simply not enough information for this approach to work with. OK. However, this remains true for the second case – the algorithm basically does not resolve the infiltration plume at all (3d-f). In the third case (3g-i), the algorithm actually outputs an infiltration plume - which could be expected since the input in this case is to first order approaching a two-layered system and no longer includes a water table below. However, and this is in my opinion crucial if such an approach is supposed to be used for studying infiltration experiments, the algorithm misplaces the position of the infiltration front by about a factor of two (the "true depth" of the plume is about 0.23 m judging from figure 3g, the calculated position clearly surpasses 0.4 m). If the results are aimed at "informing models of hydrologic processes" (L210), adding this information on top of the rather large water content deviations will certainly not be beneficial to the output of any model.

From the examples in figure 3, only the very last case (Figure 3 j-l) might be deemed an acceptable result, although the shape of the infiltration front and lateral expansion is still not captured (which would be important information for the hydrological model!). As stated above, this is most likely due to the fact that as the infiltration plume advances into the medium and increases in size, it resembles a much more

simple two-layered medium case – again without the presence of a water table. To give a better indication of whether this algorithm could - at least based on a numerical study - provide an output, which would be useful for studying the actual hydrologic infiltration process it would be necessary to present a detailed time-lapse assessment of how a progressing infiltration plume can be resolved in the first place. At minimum this could start from a time-lapse representation (e.g., a movie) of results with a good enough temporal resolution: E.g., of the "true" water content calculated by HYDRUS-2D on the left and the tomography result on the right – depicting the temporal evolution of both the infiltration event and the corresponding tomography result for each timestep. This could in principle then be used both for a rigorous error assessment, which is missing so far, and for discriminating periods in time during the infiltration process in which the situation is just too complex for the current tomography approach and where it deliver at least useful information. From the examples shown in the paper, I take it that first, the imaging fails completely, then the infiltration is resolved as being much faster than in reality while in the end a simpler situation is reached in which an acceptable result may be achieved: Hence this looks like there is a point where the inversion actually somehow converges towards reality which should be clearly identified and discussed. Without such an assessment, which does not only encompass comparing average water contents, I do not see much reason for trusting the results of the measurement inversions shown later. In 2019, for studying infiltration processes with GPR, a quantitative "average error of 5-10%" in water content is not enough if not at least the dynamics can be qualitatively resolved much better. In fact, it would be truly a pity if matching average water contents to within 10% would really be all that can be done with your elaborated dataset.

From the work presented here it seems clear that for studying dynamic unsaturated flow phenomena the authors should attempt to leverage much more of the information actually contained in the dataset. Information is already scarce for tomography algorithms based on surface data in stationary conditions. In such a dynamic infiltration experiment context, any viable approach will therefore have to give credit to the specific strengths of such a dataset. Getting more acceptable results may, e.g., include concurrently considering information from the air/groundwave and the wetting front reflection – which would likely not be directly possible in the framework of the present version of the inversion algorithm. I would also encourage the authors to take another look at the dynamics of the wetting front reflection for a source of additional information.

For getting better results by adapting the currently employed algorithm, an approach could be to constrain the inversion based on CMPs acquired at a specific time by the results from previous and subsequent time steps. Basically: If timelapse movies helped in visual interpretation of the dataset – there is no reason to expect that this will not also be the case for an automated evaluation… In my opinion, the fundamental limitation in the case presented here is not so much in the information content of the data set in itself (as stated in L.205), but in the limitations of the algorithm which would have to be discussed in a lot more detail in this paper to warrant a publication. The author's claim that "automated high-speed GPR data acquisition coupled with reflection tomography algorithms can provide a new approach to hydrologic monitoring" – will only hold if these algorithms actually leverage the additional information contained in the temporal domain. As far as I understood the author's approach, for each example shown, the pertaining spatially distributed series of CMPs is inverted without taking into account the information obtained at different times. Maybe each inversion is actually starting from a

different starting model – but to what extent this is actually the case is not clear to me from the paper and would warrant a whole discussion of its own, e.g.: how does the starting model evolve over the time series? How much does the final inverted velocity model differ from the respective starting model? Could the starting model be in some clever way constraint by results from a previous – or in an iterative approach even a subsequent – timestep? How dense would the temporal resolution have to be for such an approach to work (btw. – the inversion seems to be quite computationally intensive, which should also be discussed in terms of potential limitations: how dense could such a temporal sampling from a computational point of view actually be?

In conclusion, so far I do not see enough evidence in the paper presented here to sustain the author's main claim that "reflection tomography in the post-migrated domain is a viable method for resolving transient soil moisture content in 2D".

Hence, which way forward? I do see two possible roads to follow:

- Since the main claim can so far not be sustained, the only reason for publishing this paper would be to provide a much more thorough performance assessment of the employed algorithm under such dynamic infiltration conditions. Hence, radically refocus the publication to concentrate on assessing the true capabilities of the present algorithm under dynamic conditions based on (potentially a series of additional) numerical simulations – including some sort of time-lapse analysis /movies etc. as hinted at above. Improve on constraining the starting model and discuss in the framework of a rigorous error assessment. As stated above: Deriving average water content error is just a small part of the task if this is to be useful for studying dynamic cases. Correctly resolving the position of infiltration-induced interfaces over time is another. Water balance would be yet another – e.g., to what extent is the total amount of infiltrated water actually retrieved?
- Otherwise I would advice to keep this publication as is in the status of a discussion paper and focus the efforts on a larger inversion framework in which the present results can be one source of information, to be augmented by evaluating different aspects of the dataset. Please leverage much more of the information contained in the temporal nature of this great dataset.

In light of my rather substantial objections to publishing the current manuscript, I will not continue adding additional minor comments at this point.

---

## Referee Report (RR2)

General comments on "Reflection tomography of time-lapse GPR data for studying dynamic unsaturated flow phenomena"

Characterization of unsaturated soil water flow at different scales is essential to understand the underlying mechanisms. The potential capability of GPR has been demonstrated in a variety of studies. Identification of transient water flow using GPR is relatively easy, while proper quantification of the variability of water content in time and space is still challenging for GPR in the post-migrated domain. This study presents an efficient monitoring system to characterize sub-meter scale heterogeneous flow in a sand tank. Generally, this study is well written and presented. However, the capability of the proposed approach for quantitatively monitoring transient flow at sub-meter scale is still lack of persuasion, according to the results from the synthetic and laboratory studies. Nevertheless, this technique is promising to capture some very transient water flow peaks in space, in particular in structural soils, while it is difficult for common point measurements.

  1. The authors demonstrate the coupling automated GPR data collection with reflection tomography in synthetic studies and laboratory studies. Given a perfect hydrological model for a homogenous soil, the considerable discrepancy (5~10%) between the true water content and the estimates from reflection tomography indicates the proposed approach is not ready for hydrological applications. Further analysis of the accuracy of the tomography approach is needed.
  2. Provided such an irrigation setup for surface infiltration, heterogeneous water flow could be expected. I am wondering how these small-scale heterogeneities within a CMP gather influences on the accuracy of the reflection tomography algorithm. Please clarify this.
  3. The error (5~10%) in the synthetic studies mainly comes from the artifacts, while the serious error (5~15%) in the laboratory studies might be from improper probe locations. Concerning the foreseeable heterogeneous water flow, the comparison on the reflection tomography estimates with the probes half-meter away might not make sense. Hence, more solid validation is required to consolidate the quantitative characterization of dynamic unsaturated flow phenomena. Finally, the relationships between the two error levels (5~10% vs. 5~15%) should be discussed.
  4. The authors just demonstrate the discrepancies between reflection tomography and Probes for three-time slides (0, 95 and 173). Considering the fast evolution of the heterogeneous wetting, I am wondering how the discrepancies evolve.

**Mineral comments:**

(1) L40-41: I didn't find a multi-offset survey for infiltration experiment in Gerhards

(2008). Besides, the journal name is missing in the reference.

(2) L124: 'mS/m' to 'mS m$^{-1}$'.

(3) L134: 'Reflection TOmography Of simulations' to 'Reflection Tomography of Simulations'

(4) Line 219: The format of paper title should be just capitalized the first word. Same issue for other references.

---

## Referee Report (RR3)

[referee-annotated manuscript omitted]

---

## Author Response (AR2)

Author Response to Report #2 Submitted by Anonymous Referee #3 in Support of the Editor's Decision for manuscript # **hess-2018-230** titled:

**Mangel et al: Reflection tomography of time-lapse GPR data for studying dynamic unsaturated flow phenomena**

To the Editor:

Thank you for all your work in soliciting reviews for our manuscript and managing the submission process. Our response to points made by Anonymous Reviewer #3 are below and placed within the original text for context. Original text from the reviewer is denoted below in *italicized text*, whereas our responses are in regular font.

Overall, the authors strongly disagree with the comments of Reviewer #3. We point to what seems to be a fundamental misrepresentation (and perhaps misunderstanding) of our goals and outcomes by the reviewer. The work presented in this manuscript is the first time that time-lapse monitoring of infiltration has been attempted using GPR reflection tomography. Such work was not possible in the past due to a lack of automated data collection systems that could collect the massive volumes of data required quickly enough to enable the imaging of dynamic events, though there has been a limited history in the literature where reflection tomography has been used to image static spatial variations in water content (as we stated clearly in the manuscript). Now that we have developed an automated GPR system, dynamic imaging has become a possibility and it is of importance to report to the hydrologic (and geophysics) community the feasibility of such an approach. We note that this application of reflection tomography is distinct from static imaging in that water content distributions observed during flow may be significantly different than under static conditions (e.g., due to the possible presence of flow fronts, gradients, instabilities, fast transients, and heterogeneities impacting water content distributions). Thus our goal in this paper is to contribute a baseline understanding of this new monitoring technique for the community by "evaluat[ing] reflection tomography of high-resolution GPR data as a tool for observing and characterizing unsaturated flow patterns during infiltration" (lines 58-59). We perform this evaluation using accepted reflection tomography algorithms that represent the state-of-the-science and we report our findings clearly and directly, i.e., in the manner that is required for the integrity and progress of science, not in a manner that simply has the objective of publishing a "nice result". To summarize, our manuscript makes the following important contributions:

(1) this first of its kind study establishes that time-lapse GPR reflection tomography of a dynamic infiltration event is now technically possible, and we set a reference point for future studies;
(2) we use numerical modeling to establish a baseline for the potential and limitations of time-lapse GPR reflection tomography under idealized conditions – it is not possible for a real-world application to perform better than what is shown in the synthetic study (Figure 3) and the scenarios given provide insight regarding where GPR reflection tomography is likely to fail in practice;
(3) we provide estimates of the quantitative error to be expected in an imaging scenario utilizing the current state-of-the-science reflection tomography techniques (these errors will obviously decrease in future as new inversion algorithms are developed for dynamic imaging problems); and
(4) we show that it is possible to achieve an imaging accuracy with real empirical data that is similar to that achieved with the synthetic baseline (i.e., the limitations are in the inversion algorithm, not the data).

We contend that these findings are important outcomes to report to the scientific community as they establish a scientifically sound and robust assessment of this new method's potential and limitations when used in hydrological applications.

Reviewer #3 did not present any evidence in their comments arguing that similar work has been previously reported in the literature or that there are any scientific flaws in our analysis that could potentially be corrected or provide justification for rejection of the manuscript. The fact that the reviewer does not like the results that we reported, which are based on a well-established and technically sound analysis approach that is well-documented in the literature, is neither relevant nor reflective of the scientific method: the results we obtained are what they are and it is our duty to report these results to the community. The fact that the reviewer states that they see value in having our work remain available to the community as a "discussion paper" is clear evidence of their perceived value of the manuscript and contradictory to their recommendation that the paper be rejected.

While we appreciate and agree with the reviewer's enthusiasm for the potential of GPR to provide additional and deeper insights into unsaturated flow, the arguments laid out by the reviewer below are the framings of a future research program, not critical assessments of our analysis or the results that we have presented. We agree that there are opportunities to develop new algorithms and approaches that could indeed provide improvements over the existing and accepted reflection tomography methodology. These new algorithms do not yet exist, however, and it is therefore completely unreasonable for a reviewer to suggest that our paper should be rejected because we did not use such non-existent analysis strategies, particularly when our work has been based on sound science and has been recognized as a valuable contribution by multiple reviewers.

We are hopeful that the two previous critical reviews, which the authors have addressed, and the current recommendation by a third reviewer to publish the manuscript as-is, are sufficient to justify publication of this manuscript in HESS. We are prepared to make further revisions to the manuscript given guidance, but the review below does not provide actionable items justifying changes at this time. Further responses to each reviewer comment are given below.
* * *
*The paper under review for publication in HESS by Mangel et al aims at employing a reflection tomography based inversion algorithm, which is well-established for calculating subsurface velocity distributions from CMP GPR measurements in stationary conditions for deriving – by proxy – subsurface water content – distributions. In contrast to previous publications, here the focus is on dynamically changing conditions during infiltration experiments.*

*First of all, I would like to specifically laud the authors for their dedicated experimental approach and congratulate them for their laboratory setup and the undoubtedly involved data set which may yet hold the key to studying the infiltration experiments they monitored by GPR in so much detail. However, the key question for whether the currently submitted work warrants a dedicated publication is whether the authors found a novel and robust way to extract meaningful and relevant information from this great dataset. Unfortunately, I am convinced that the inversion approach chosen for this publication in its current form falls short of achieving that aim (i.e., as the title states: Usage of this algorithm for "studying dynamic unsaturated flow phenomena") and does not give justice to the information potentially contained in their elaborated dataset.*

The authors appreciate praise in performing the research but argue that the data presented does indeed represent "a novel and robust way to extract meaningful and relevant information" given that currently there are no other methods (geophysical or otherwise) to completely non-invasively monitor water content of soils at this resolution over large areas. If the reviewer feels that there are comparable methods that demonstrate our work does not represent an advance in the field, they should have cited that literature to support their argument. There failure to do so leaves us unable to respond and we believe this is a result of the fact that no such prior work that exists in the literature given that ours is a first-of-its-kind study. While we agree with the reviewer that the errors are high compared to what can be achieved with a point sensor (e.g., TDR), there is no comparable baseline for a completely non-invasive dynamic GPR imaging technique such as that obtained here (e.g., Moysey et al. (2004) showed that even borehole-based GPR tomography, which is a much simpler and better constrained problem than that of surface-based reflection tomography, can have errors >10% (vol./vol.) water content simply due to inversion errors and scale).

Furthermore, our goal here was to test and evaluate the existing reflection tomography algorithm and evaluate its performance, not to develop a new algorithm. The "novelty" in this work is in taking a unique dynamic data set, which has until now been unachievable, and combining it with a standard, yet state-of-the-science algorithm for analyzing the data. We welcome the opportunity to share this data set with others through this publication in HESS, which will allow for the development and testing of new algorithms that can be compared against the baseline performance established in this manuscript.

*The inversion algorithm's trouble is quite clearly shown already by the simulation based results the authors present in Figure 3: Here, the authors first calculate water content distributions from HYDRUS-2D (figure 3, left column), then derive GPR profiles from these distributions (examples shown in figure 2) and feed these into their tomography algorithm to retrieve the respective water content distributions (figure 3, center column). In the first case, as the authors admit themselves, their algorithm fails completely to capture the velocity profile, since there is simply not enough information for this approach to work with. OK.*

Note that we revise figure 3 as a result of a later comment by Reviewer #3, though it does not alter this discussion point. The authors agree that the results presented in figure 3a-c leave much to be desired. We discussed this in the manuscript as stemming from an intrinsic limitation of the information in the data due to a lack of GPR reflectors: only the average water content can be determined because of the limited spatial data coverage near the reflection targets (i.e., at the bottom interface of the model) and this is therefore an intrinsic limitation of the data and not a critical limitation of the algorithm itself (lines 152-156). The goal of the modeling is to demonstrate scenarios highlighting the strengths and weaknesses of the reflection tomography approach. Simply reproducing the water content distribution is unrealistic for any tomographic method and thus we instead aspire to help the reader understand challenges that may be faced in imaging different flow scenarios, i.e., in this case the fact that the lack of reflectors prevents the estimation of vertical water content variability. Furthermore, the authors argue the significance of this result as it directly illustrates one limitation to the approach; if minimal reflectors are present in the data, the tomography results suffer. In general, it is critically important to not only understand where the method succeeds, but also where it fails. Understanding the limitations of any method is crucial to application and deserves just as much attention as more successful results.

It is possible that other algorithms could help to alleviate the limitation of the imaging to some extent. For example, it is possible that more advanced algorithms like full-waveform inversion could make use of reflection amplitudes to better account of the vertical variability, though it is not clear that this is the case given that the fundamental limitation here was the lack of reflection points due to the homogeneity of the subsurface and absence of a target infiltration plume. Full-waveform inversion is still being developed for surface GPR applications, however, and is significantly beyond the scope of this manuscript [*Ernst et al.* 2007; *Meles et al.* 2010; *Busch et al.* 2014; *Lavoué et al.* 2014]. Even if it were to be found in later studies that full waveform inversion could improve the imaging, it still requires a starting model that would likely be obtained from a reflection tomography approach like that we have presented (lines 211-213).

*However, this remains true for the second case – the algorithm basically does not resolve the infiltration plume at all (3d-f).*

This statement by the reviewer is demonstrably false as Figure 3f shows an error of only a few percent behind the wetting front. The results also illustrate that the shape of the wetting front can be reconstructed by the tomography algorithm. Comparing Figure 3f to the exact same case with no infiltration front (Figure 3c) clearly demonstrates that the additional information provided by reflections generated by the presence of the wetting front has resulted in new, spatially localized information. We highlighted this in the discussion on lines 157-162. The problem here is that the algorithm struggles in imaging the gradient nature of the capillary fringe present at the base of the model, though this limitation was already clear from the prior scenario. Perhaps the reviewer was misled by the minimal color contrast in figure 3e, which the authors can correct by adjusting the colormap if required. Regardless, no scientific metric is presented in the reviewer's assessment of the results to determine what standard the reconstruction is being held against.

We have corrected one error in the text identified on line 162 of the manuscript where Figure 2g was misidentified as Figure 2e.

*In the third case (3g-i), the algorithm actually outputs an infiltration plume - which could be expected since the input in this case is to first order approaching a two-layered system and no longer includes a water table below.*

The arguments presented by the reviewer that the system is approaching a simple two-layered system are irrelevant. The advantage of the reflection tomography algorithm over others is that it can account for lateral and vertically variable velocity fields. The same argument for a less complex 'layered system' could be made regarding the model presented in figure 3a-c, yet the tomography algorithm cannot resolve the layered structure in that case. As the authors mention, this is due to limited spatial coverage by the GPR data. Furthermore, the phrasing used here by the reviewer, e.g. 'which could be expected', is not consistent with the reality of tomographic imaging methods which often face spatial resolution, smoothness, and trade-off issues. The results observed are not unexpected and need to be addressed to avoid misinterpretation by future practitioners of GPR reflection tomography.

*However, and this is in my opinion crucial if such an approach is supposed to be used for studying infiltration experiments, the algorithm misplaces the position of the infiltration front by about a factor of two (the "true depth" of the plume is about 0.23 m judging from figure 3g, the calculated position clearly surpasses 0.4 m). If the results are aimed at "informing models of hydrologic processes" (L210), adding this information on top of the rather large water content deviations will certainly not be beneficial to the output of any model. From the examples in figure 3, only the very last case (Figure 3 j-l) might be deemed an acceptable result, although the shape of the infiltration front and lateral expansion is still not captured (which would be important information for the hydrological model!). As stated above, this is most likely due to the fact that as the infiltration plume advances into the medium and increases in size, it resembles a much more simple two-layered medium case – again without the presence of a water table.*

The reviewer raises a good point here; the 0.10 m error in the depth of the wetting front lacks context within the inversion algorithm. Therefore, the authors have added the following sentence when discussing these results.

"The tomography algorithm overestimates the depth of the wetting front by roughly 0.10 m for the case presented in Figure 3g-i, which is likely due to smoothing effects required to regularize the inversion or an error in the picking of the wetting front horizon."

*To give a better indication of whether this algorithm could - at least based on a numerical study - provide an output, which would be useful for studying the actual hydrologic infiltration process it would be necessary to present a detailed time-lapse assessment of how a progressing infiltration plume can be resolved in the first place. At minimum this could start from a time-lapse representation (e.g., a movie) of results with a good enough temporal resolution: E.g., of the "true" water content calculated by HYDRUS-2D on the left and the tomography result on the right – depicting the temporal evolution of both the infiltration event and the corresponding tomography result for each timestep. This could in principle then be used both for a rigorous error assessment, which is missing so far, and for discriminating periods in time during the infiltration process in which the situation is just too complex for the current tomography approach and where it deliver at least useful information. From the examples shown in the paper, I take it that first, the imaging fails completely, then the infiltration is resolved as being much faster than in reality while in the end a simpler situation is reached in which an acceptable result may be achieved: Hence this looks like there is a point where the inversion actually somehow converges towards reality which should be clearly identified and discussed. Without such an assessment, which does not only encompass comparing average water contents, I do not see much reason for trusting the results of the measurement inversions shown later. In 2019, for studying infiltration processes with GPR, a quantitative "average error of 5-10%" in water content is not enough if not at least the dynamics can be qualitatively resolved much better. In fact, it would be truly a pity if matching average water contents to within 10% would really be all that can be done with your elaborated dataset.*

The reviewer misunderstands the goal of the modeling.  We do not represent different times during infiltration event, but rather cases designed to illustrate different scenarios of water content distribution that can provide insights in the interpretation of time-lapse data (i.e., no infiltration plume, the influence of a capillary fringe on the imaging, and the case where no diffuse water table is present).  Regardless of the fact that we present only a few representative times for the tomographic imaging of the experimental dataset (i.e., before, during and after the infiltration event), we selected these specific times to illustrate that the reflection tomography does indeed show substantial changes in water content that are realistic and generally consistent with measurements made at point probes (at least in trend if not in magnitude due to scale issues between the imaging and point sensors).  The reviewer does not demonstrate an understanding of the intensive effort required for inversion of these data sets.  The work for this manuscript was performed in a seismic imaging software which is engineered to handle large seismic data sets, similar to those obtained here, and involves substantial manual intervention to identify reflections in the data.  Other limitations include the use of Kirchoff migration, which the authors note as a limitation in migrating the data (lines 208-211).  We agree that future efforts building on this manuscript should involve the development of algorithms that can facilitate the analysis of massive timelapse datasets like those we can now collect.

Finally, the results we report are not a matter of "trust".  They are the result of direct comparisons between the imaging results and point sensors (or true model for the synthetic data).  Such comparisons and the reporting of these results are the foundations upon which science is built and advanced through successive improvements and discoveries over time.  Thus the results we report here are required as a foundation upon which future work can be built and compared to demonstrate further advances.  It appears that this reviewer does not understand this fundamental premise of science.  Instead it appears that the reviewer favors tweaking algorithms to cherry pick "good" results, which we believe to be an increasingly common and dangerous approach to science.  We fundamentally reject this philosophy and hope that the editors of HESS agree that such an approach is not an approach that promotes the continued development of strong scientific community.

*From the work presented here it seems clear that for studying dynamic unsaturated flow phenomena the authors should attempt to leverage much more of the information actually contained in the dataset. Information is already scarce for tomography algorithms based on surface data in stationary conditions. In such a dynamic infiltration experiment context, any viable approach will therefore have to give credit to the specific strengths of such a dataset. Getting more acceptable results may, e.g., include concurrently considering information from the air/groundwave and the wetting front reflection – which would likely not be directly possible in the framework of the present version of the inversion algorithm. I would also encourage the authors to take another look at the dynamics of the wetting front reflection for a source of additional information.*

The authors are uncertain on the reviewer's use of '*much more of the information actually contained in the dataset*'. The reviewer mentions analysis of the air wave and groundwave in the data. However, the airwave contains absolutely no information regarding the infiltration process. The groundwave cannot be used in reflection tomography because it is not a reflected arrival, but a direct arrival. Analysis of the groundwave analysis could be included in future efforts but would only help to constrain the results in the upper ~0.10 m of the inversion, thus it is not clear how that suggestion is particularly helpful.

We again agree with Reviewer #3 that there is indeed much potential for future studies advancing and refining algorithms for time-lapse reflection tomography, but once again we emphasize that this was not the goal that we have identified for this manuscript and well beyond the scope of work here.

*For getting better results by adapting the currently employed algorithm, an approach could be to constrain the inversion based on CMPs acquired at a specific time by the results from previous and subsequent time steps. Basically: If timelapse movies helped in visual interpretation of the dataset –there is no reason to expect that this will not also be the case for an automated evaluation…*

*In my opinion, the fundamental limitation in the case presented here is not so much in the information content of the data set in itself (as stated in L.205), but in the limitations of the algorithm which would have to be discussed in a lot more detail in this paper to warrant a publication. The author's claim that "automated high-speed GPR data acquisition coupled with reflection tomography algorithms can provide a new approach to hydrologic monitoring" – will only hold if these algorithms actually leverage the additional information contained in the temporal domain. As far as I understood the author's approach, for each example shown, the pertaining spatially distributed series of CMPs is inverted without taking into account the information obtained at different times. Maybe each inversion is actually starting from a different starting model – but to what extent this is actually the case is not clear to me from the paper and would warrant a whole discussion of its own, e.g.: how does the starting model evolve over the time series? How much does the final inverted velocity model differ from the respective starting model? Could the starting model be in some clever way constraint by results from a previous – or in an iterative approach even a subsequent – timestep? How dense would the temporal resolution have to be for such an approach to work (btw. – the inversion seems to be quite computationally intensive, which should also be discussed in terms of potential limitations: how dense could such a temporal sampling from a computational point of view actually be?*

[Figure]

Figure 1: Progression from starting model to inversion results for selected numerical simulations.

The authors explicitly mention how the starting models were derived for the tomography algorithm (lines 87-88). In Figure 1 of this response, we show the starting models derived from semblance analysis of the simulations. We have replaced Figure 3 in the manuscript with this figure and edited the appropriate references in order to show the initial models for the tomography algorithm.

Again, we agree that there is potential for developing new algorithms for the inversion of time-lapse reflection tomography data and we are (of course) working on developing these tools, though again we emphasize that they currently do not exist.

*In conclusion, so far I do not see enough evidence in the paper presented here to sustain the author's main claim that "reflection tomography in the post-migrated domain is a viable method for resolving transient soil moisture content in 2D".*

*Hence, which way forward? I do see two possible roads to follow:*
- *Since the main claim can so far not be sustained, the only reason for publishing this paper would be to provide a much more thorough performance assessment of the employed algorithm under such dynamic infiltration conditions. Hence, radically refocus the publication to concentrate on assessing the true capabilities of the present algorithm under dynamic conditions based on (potentially a series of additional) numerical simulations – including some sort of time-lapse analysis /movies etc. as hinted at above. Improve on constraining the starting model and discuss in the framework of a rigorous error assessment. As stated above: Deriving average water content error is just a small part of the task if this is to be useful for studying dynamic cases. Correctly resolving the position of infiltration-induced interfaces over time is another. Water balance would be yet another – e.g., to what extent is the total amount of infiltrated water actually retrieved?*
- *Otherwise I would advice to keep this publication as is in the status of a discussion paper and focus the efforts on a larger inversion framework in which the present results can be one source of*

*information, to be augmented by evaluating different aspects of the dataset. Please leverage much more of the information contained in the temporal nature of this great dataset. In light of my rather substantial objections to publishing the current manuscript, I will not continue adding additional minor comments at this point.*

The authors respectfully disagree with the reviewer based on all of the previous replies above and the fact that three other reviews did not agree with the conclusions of this reviewer. By keeping this publication "in the status of a discussion paper" it seems that the reviewer is suggesting that the content here should remain available to the scientific community. If this is the case, then even this reviewer does indeed see value in having our work available to the public and reinforces the fact that this work should indeed be published.

[revised manuscript text omitted]

---

## Author Response (AR3)

This response is directed toward the comments left by Anonymous Referee #4, posted to the Hydrology and Earth Systems Science (HESS) discussion board for manuscript hess-2018-230 on August 28, 2019.

The authors would like to thank the reviewer for devoting time to reviewing our manuscript and for expeditiously providing a critical review of the content. Below, the authors have outlined responses to individual comments made by the reviewer.
* * *
*AR4: The authors demonstrate the coupling automated GPR data collection with reflection tomography in synthetic studies and laboratory studies. Given a perfect hydrological model for a homogenous soil, the considerable discrepancy (5~10%) between the true water content and the estimates from reflection tomography indicates the proposed approach is not ready for hydrological applications. Further analysis of the accuracy of the tomography approach is needed.*

**Author Response:** This is the first of its kind study which establishes that time-lapse GPR reflection tomography of a dynamic infiltration event is now technically possible, thus establishing a reference point for future studies. The study points out shortcomings of the state-of-the-art in reflection tomography and offers potential solutions which are best explored in detail in later works that can focus on examining these issues. The authors maintain that the findings presented here are important to the hydrogeophysics community as they establish a scientifically sound and robust assessment of the method's potential and limitations when use in hydrological applications.

*AR4: Provided such an irrigation setup for surface infiltration, heterogeneous water flow could be expected. I am wondering how these small-scale heterogeneities within a CMP gather influences on the accuracy of the reflection tomography algorithm. Please clarify this.*

**Author Response:** As with any measurement of soil moisture, there are limitations regarding the scale at which heterogeneities can be resolved. For example, the moisture probes used in this work are averaging the volumetric moisture content over a cylindrical volume of roughly 143 cubic centimeters (Meter Group). Similarly, with GPR methods it is generally accepted that the resolution of the GPR signal is roughly 25% of the wavelength of the signal (see figure 2 of Mangel et al. 2015). The wavelength of the signal for this work is roughly 10 centimeters (varies with velocity/water content). Small-scale heterogeneities are accounted for in this work by using a ray tracing algorithm capable of accounting for these small-scale heterogeneities (curved ray tracer) in the pre-stack depth migration. The dense spacing of the CMPs at 0.2 m also provides repeated measurements of the same volume, increasing the ability of the data to resolve these small changes in soil moisture.

*AR4: The error (5~10%) in the synthetic studies mainly comes from the artifacts, while the serious error (5~15%) in the laboratory studies might be from improper probe locations. Concerning the foreseeable heterogeneous water flow, the comparison on the reflection tomography estimates with the probes half-meter away might not make sense. Hence, more solid validation is required to consolidate the quantitative characterization of dynamic*

*unsaturated flow phenomena. Finally, the relationships between the two error levels (5~10% vs. 5~15%) should be discussed.*

**Author Response:** We agree with the reviewer that it in an ideal scenario, the moisture probes would be located in the plane of the GPR image and that this offset could be contributing to the observed discrepancies between probes and GPR for the experimental case given the heterogeneous flow that occurred. The moisture probes were located off the GPR transect to avoid backscattering in the transmitted signal that would interfere with analysis of the GPR wave velocities. Though this is not an ideal setup for comparison of the moisture probe data to the GPR estimated of water content, it was necessary to ensure that high-quality reflection tomography data could be collected. We have modified the conclusions to emphasize this point as a potential source of error in the experiment at line 204-205 of the manuscript.

Notably this is not an issue for the numerical experiment, so does not fully explain all of the error in the water content estimates. Therefore, the two main sources of discrepancies between the water content probes and tomography results are likely (i) the lack of a coherent wetting front reflection as seen in numerical experiments reducing the available data for the reflection tomography (i.e., if more reflectors were present, an improved image would like result), and (ii) experimental errors that include discrepancy in effective measurement scale between the inversion and probes, consequences of out of plane probes, or other general experimental errors. We have expanded the discussion on lines 201-209 to elaborate on these issues.

*AR4: The authors just demonstrate the discrepancies between reflection tomography and Probes for three-time slides (0, 95 and 173). Considering the fast evolution of the heterogeneous wetting, I am wondering how the discrepancies evolve.*

**Author Response:** We have addressed a similar comment in earlier reviews of the manuscript.

The authors selected these specific times to illustrate that the reflection tomography does indeed show substantial changes in the water content that are realistic and generally consistent with measurements made at point probes; at least in trend if not in magnitude due to scale issues between the imaging and point sensors.

The work for this manuscript was performed in a seismic imaging software which is engineered to handle large static seismic data sets and involves substantial manual intervention to identify reflections in the data. Processing the data for each time is therefore a substantial burden and we do not have results available from additional times. Furthermore, the software is unable to account for electrical conductivity of the soil and utilizes Kirchoff migration, which the authors identify as a limitation in properly migrating the data.

Despite the fact that of the presented tomography results in time, the selected times greatly enhance the information content of the infiltration process in the spatial domain compared to invasive methods (i.e., probes). The authors plan to continue with this research once a more accessible and hydrogeophysically-oriented software is developed by the lead author, which will substantially increase the efficiency and ability to perform these analyses at a higher temporal discretization, but this is well beyond the scope of this manuscript.

**Mineral comments:**

*AR4: L40-41: I didn't find a multi-offset survey for infiltration experiment in Gerhards (2008). Besides, the journal name is missing in the reference.*

**Author Response:** Figure 1 of the Gerhards et al (2008) paper shows GPR acquisition setup. From the figure, it can be seen that data are being collected at multiple antenna offsets, which they refer to as multi-channel.

The journal for the reference is Geophysics, which is properly cited in the bibliography on line 245 of the PDF version of the second revision of the submitted manuscript (uploaded Feb 28, 2019).

*AR4: L124: 'mS/m' to 'mS m-1'.*

**Author Response:** The change has been incorporated into the new revision of the manuscript.

*AR4: L134: 'Reflection TOmography Of simulations' to 'Reflection Tomography of Simulations'*

**Author Response:** The heading for section 3 of the manuscript is correct in the PDF version of the second revision of the submitted manuscript (uploaded Feb 28, 2019).

*AR4: Line 219: The format of paper title should be just capitalized the first word. Same issue for other references.*

**Author Response:** The change has been incorporated in the new revision of the manuscript.

Author Response to Interactive Comment on "Reflection tomography of time-lapse GPR data for studying dynamic unsaturated flow phenomena" by Adam R. Mangel et al.

This response is directed toward the comments left by Referee #5, Timothy Bechtel, posted to the Hydrology and Earth Systems Science (HESS) discussion board for manuscript hess-2018-230 on October 16, 2019.

The authors would like to thank the reviewer for devoting time to reviewing our manuscript and for expeditiously providing a critical review of the content. The authors also recognize the reviewer for identifying themselves as they view this as a step forward in a truly open manuscript review process. Below, the authors have outlined responses to individual comments made by the reviewer.

We have accepted all typographic suggestions made by the reviewer and incorporated them into the revision of the manuscript.

[revised manuscript text omitted]